# Optimizing spectral quality with quantum dots to enhance crop yield in controlled environments

Charles H. Parrish II [1], Damon Hebert[2], Aaron Jackson[2], Karthik Ramasamy[2], Hunter McDaniel [2], Gene A. Giacomelli [1✉] & Matthew R. Bergren [2✉]

Bioregenerative life-support systems (BLSS) involving plants will be required to realize self-sustaining human settlements beyond Earth. To improve plant productivity in BLSS, the quality of the solar spectrum can be modified by lightweight, luminescent films. $CuInS_2/ZnS$ quantum dot (QD) films were used to down-convert ultraviolet/blue photons to red emissions centered at 600 and 660 nm, resulting in increased biomass accumulation in red romaine lettuce. All plant growth parameters, except for spectral quality, were uniform across three production environments. Lettuce grown under the 600 and 660 nm-emitting QD films respectively increased edible dry mass (13 and 9%), edible fresh mass (11% each), and total leaf area (8 and 13%) compared with under a control film containing no QDs. Spectral modifications by the luminescent QD films improved photosynthetic efficiency in lettuce and could enhance productivity in greenhouses on Earth, or in space where, further conversion is expected from greater availability of ultraviolet photons.

[1] Controlled Environment Agriculture Center, The University of Arizona, Tucson, AZ 85719, USA. [2] UbiQD, Inc., Los Alamos, NM 87544, USA. ✉email: giacomel@ag.arizona.edu; matt@ubiqd.com

As anthropogenic climate change exacerbates the severity and frequency of extreme weather events[1], arable farmland is decreasing due to overpopulation and drought[2]. For resilience against these threats, producers of vegetable crops have turned to controlled environment agriculture (CEA) as a solution. CEA affords crop production with the highest resource use efficiency (RUE), and exceeds the annual per-area yield of field agriculture tenfold[3]. This greater productivity results from optimized environmental controls, year-round production, and reduced pests and diseases. Furthermore, recirculating hydroponic plant nutrient delivery systems recycle water for greatest RUE[4].

The 400–700 nm waveband of the electromagnetic spectrum primarily powering photosynthesis is referred to as photosynthetically active radiation (PAR). The amount of PAR photons a plant receives is quantified by the photosynthetic photon flux density (PPFD, $\mu mol\ m^{-2}\ s^{-1}$), which is the number of photosynthetically active photons per unit area per second. The PPFD a plant receives in a 24 h day is measured as the daily light integral (DLI, $mol\ m^{-2}\ d^{-1}$) and is directly proportional to the rate of plant growth, which is subsequently determined as biomass accumulation. While PPFD and DLI are important plant growth parameters for characterizing plant development, they ignore spectral quality differences of radiation received by the plant.

Spectral quality is an important growth parameter since the efficiency of photosynthesis is dependent on the wavelength of the photon absorbed by the plant. The wavelength-dependent photosynthetic efficiency can be determined by the photosynthetic quantum yield (QY), a measure of the production of oxygen or consumption of carbon dioxide, for various crops[5,6]. The photosynthetic action spectrum compares the relative photosynthetic QY per wavelength, and indicates that wavelengths between 575 and 675 nm are ~30% more efficient for photosynthesis than photons in the blue waveband (400–500 nm).

Several recent studies about light spectra effects on plant growth have determined that both light quantity and quality affected plant morphology[7–10]. Using a broad-spectrum light source to grow plants outperformed monochromatic light treatments, but modifying the relative percentages of different wavelengths produced varied and improved results. For example, the sensitivity of seven plant species to blue and green light with a daylight-equivalent $500\ \mu mol\ m^{-2}\ s^{-1}$ PPFD reduced dry mass in tomatoes, cucumbers, and peppers when blue light was increased between 11% and 28% of the total spectrum[8]. At $200\ \mu mol\ m^{-2}\ s^{-1}$ PPFD, tomatoes continued to be negatively affected, where dry mass was reduced by 41%, while no difference was observed in peppers or cucumbers. These results illustrated the importance of spectral quality for influencing plant response, particularly in high light intensity environments, and suggest that there are optimal spectra for different plant species.

CEA food production is particularly advantageous in extreme environments, like outer space, where regular access to fresh vegetables is impractical. NASA and other space agencies are developing CEA systems for crewed space applications[11,12], which range from growth chambers with light-emitting diode (LED)-based lighting to drive photosynthesis[13], to deployable greenhouses that could utilize sunlight for photosynthesis, like the Prototype Mars-Lunar Greenhouse (MLGH)[14].

Until recently, only limited options have been available commercially for controlling the quality of light for greenhouse crop production besides expensive and energy-intensive electrical lighting[15]. Furthermore, artificial lighting requires relatively heavy electrical equipment to deploy, operate, and maintain, which limits applications for extended space missions. Photoselective films, which filter out certain wavelengths have been used to modify the solar spectrum, but these materials always reduce light intensity, and may not be suitable for light-limited geographic locations[16,17]. Therefore, technologies that improve light quality without increasing energy demand, excessive physical mass, deployment challenges, or expense, would be beneficial for terrestrial horticulture and are required for space applications where these constraints are even greater. Previous work on luminescent agriculture films have used organic fluorophores[18] or other fluorescent materials[19] to modify the solar spectrum without electricity, however these materials have been limited to only specific wavelength emissions, have poor photon conversion efficiencies, and/or suffer from stability issues. Therefore, quantum dots should be considered as an alternative fluorescent material due to their high photon conversion efficiency, tunable emission spectrum and improved stability compared to fluorescent organic dyes.

Quantum dots (QDs) have been utilized for a variety of applied technologies, including remote phosphors for displays[20,21], solid-state lighting[22,23], solar harvesting electricity generation[24,25], cancer detection[26–28], and even for studying pollination habits of bees[29,30]. In this study, $CuInS_2/ZnS$ (CIS/ZnS) QDs are incorporated into a film to down-convert UV and blue photons to orange and red photons. Compared with other QD compositions, CIS/ZnS QDs have unique optical properties: a size-tunable PL emission, a very high PL QY, and a broad-spectrum emission[28]. For QDs in general, a broad emission is generally associated with a dispersion of QD sizes or compositions in an ensemble and typically result in lower conversion efficiencies due to an unoptimized synthesis procedure, but for CIS/ZnS this is a result of a defect-mediated emission[31]. Additionally, CIS/ZnS QDs exhibit a large Stokes shift, allowing for absorption in the UV and blue (see Supplementary Figs. 18, 19) while emitting in the red or orange, which minimizes PAR self-absorption and shifts the spectrum to more photosynthetically efficient wavelengths.

Here, an approach is presented that incorporates luminescent CIS/ZnS QDs into flexible luminescent agriculture films to passively modify the solar spectrum. Repeated experiments on Lactuca sativa L. (cv. "Outredgeous" red romaine lettuce) were conducted within a semi-closed plant growth system including QD films, which increased edible dry and fresh biomass as well as total leaf area compared with a control film. These modified spectra improved light quality for photosynthesis and thereby enhanced the photosynthetic efficiency and productivity of crops grown in CEA.

## Results and discussion

**Spectral control with CIS/ZnS QD films.** Two different QD films were manufactured, with peak emissions centered at 600 nm (orange, O-QD) and 660 nm (red, R-QD). These wavelengths were chosen due to the strong overlap between their emission spectra and the wavelength range associated with the highest relative photosynthetic QY for lettuce (Fig. 1a).

To evaluate how the modified spectra from the QD films affect plant growth, indoor plant trials were conducted with red romaine lettuce (Lactuca sativa L. cv. "Outredgeous") in a custom built, plant growth test chamber (PGTC). The PGTC was designed to maintain a uniform environment across all plant growing areas, two of which were under O-QD and R-QD films and the third was under a Control (C) polyethylene film without QDs (Fig. 1b).

The total PFD and the spectral distribution within each light treatment zone is shown in Table 1. PPFD values were measured with an Apogee quantum sensor and refer only to PAR (B, G, and R), while percentages shown in Table 1 are relative to the entire PFD (300–800 nm) range spanning UV, PAR, and FR as measured with an Apogee spectroradiometer. The ultraviolet (UV) and blue

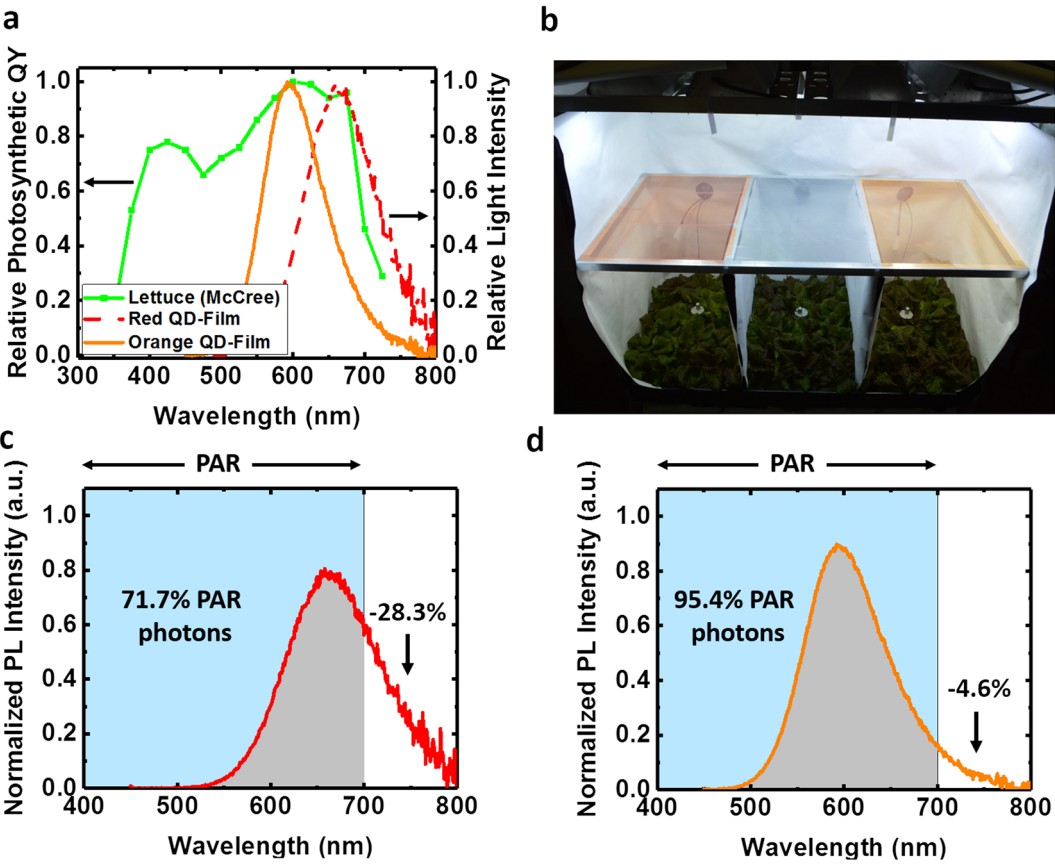

**Fig. 1 Luminescent properties of QD films. a** The emission spectra of the two QD films with peak PL emissions of 600 nm (orange, solid line) and 660 nm (red, dashed line) overlapped on the photosynthetic response curve for lettuce (green, squares) measured by McCree[5]. **b** Photograph of the PGTC for lettuce plant trials. Three light treatment areas were used to compare different light spectra created by QD films (left and right) to a Control film (middle) under similar light intensity. The spectra of QD films with peak emission of (**c**) 660 nm and (**d**) 600 nm were normalized to have an area of 1 and then integrated over PAR wavelengths (400–700 nm) to determine the portion of the emission that was outside of PAR.

**Table 1 Characterization of light intensities and spectral distribution from 300 to 800 nm under QD and Control films in the PGTC prior to experimentation.**

|  | Control film | O-QD film | R-QD film |
|---|---|---|---|
| PFD: 300–800 nm [μmol m⁻² s⁻¹] | 435 | 458 | 413 |
| PPFD: 400–700 nm [μmol m⁻² s⁻¹] | 382 (86.1%) | 395 (85.4%) | 338 (80.3%) |
| UV: 300–400 nm [μmol m⁻² s⁻¹] | 9 (2.1%) | 7 (1.4%) | 4 (0.9%) |
| Blue: 400–500 nm [μmol m⁻² s⁻¹] | 136 (31.3%) | 107 (23.3%) | 85 (20.5%) |
| Green: 500–600 nm [μmol m⁻² s⁻¹] | 156 (35.8%) | 166 (36.4%) | 133 (32.3%) |
| Red: 600–700 nm [μmol m⁻² s⁻¹] | 90 (20.8%) | 122 (26.7%) | 120 (29.1%) |
| Far-red: 700–800 nm [μmol m⁻² s⁻¹] | 44 (10.2%) | 56 (12.1%) | 71 (17.2%) |
| R:FR | 2.1 | 2.2 | 1.7 |

Percentages listed refer to % of PFD.

(B) wavelengths were clearly reduced under both of the QD films compared with the Control, while the intensities of O, R, and far-red (FR) were increased compared with the Control.

With an Apogee Instruments PS-300 spectroradiometer, spectral quality was measured before and between repeated experiments to evaluate changes in film performance throughout the experiment. Film performance remained stable throughout the study, where the statistically significant spectral shift for both QD-film treatments compared to the control treatment were equivalent after each experiment (see Supplementary Fig. 20).

To ensure differences in production solely resulted from spectral differences due to the QD films, the light intensity incident upon each film was ensured to be uniform (see Supplementary Figs. 1–4 and Supplementary Table 1 for more information). While the difference between PPFD values measured under the C and O-QD films were within 3%, PPFD under the R-QD film the PPFD was 12% lower than that of the control film. This lower value was attributed to a portion of the R-QD emission extending beyond 700 nm, which is outside the PAR range, and therefore was not measured by the quantum sensor. This is illustrated in Fig. 1c, d, where normalized emission spectra of the R-QD and O-QD films, respectively, indicated only 4.6% of the emission spectrum from O-QD film was beyond PAR (>700 nm), but 28.3% of the emission from the R-QD film did not contribute to PPFD. Therefore, even if the total number of incident photons was held constant for the three treatment areas, the measured PPFD incident upon the plants is always lower for the R-QD film compared to the other light treatments. The concentration of QDs in the O-QD and the R-QD films were similar and designed to ensure similar overall absorption of the metal-halide (MH) light source. Since both films had similar absorption and had equivalent PL QYs of ~85%, equivalent quantities of photons were converted by each film.

Additionally, attempting to increase the light intensity above the R-QD film to ensure similar PPFD below the film could

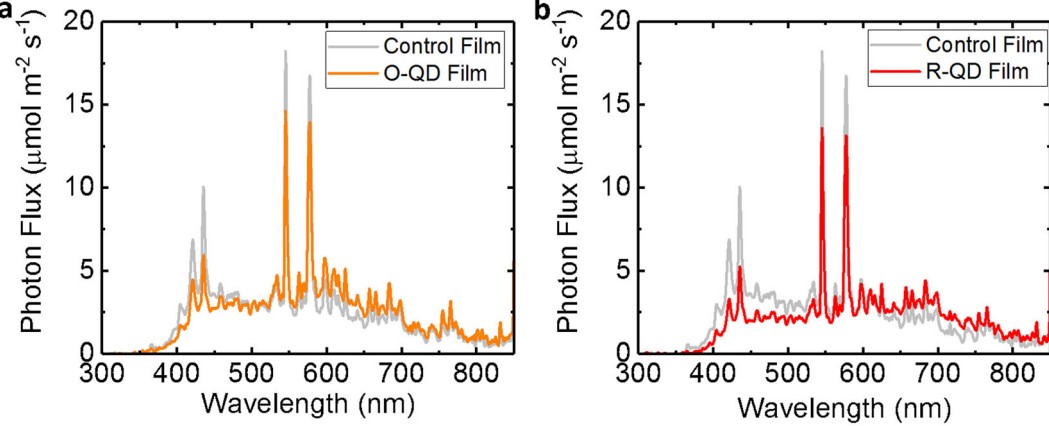

**Fig. 2 Measured spectra for each of the three light treatments. a** Measured spectra beneath the O-QD film and under the control film. **b** Measured spectra beneath the R-QD film and control film.

have provided an additional benefit to the plants due to an accompanying increase of FR photons, which have been shown to equally drive photosynthesis when acting synergistically with existing PAR photons[32].

At the time of film manufacture, PPFD spectra were measured underneath the O-QD, R-QD, and Control films with a spectrophotometer (Optimum Optoelectronics Corp., SRI-PL-6000), and comparisons of spectra between the QD films and Control film are shown in Fig. 2.

**Spectral effects on red romaine lettuce production**. The plant study consisted of three, 28-day experiments each consisting of 36 red romaine lettuce plants ($N = 36$) divided equally among three experimental treatments ($n = 12$). Plants were harvested 28 days after sowing (DAS). Postharvest measurements included total leaf area (TLA, $cm^2$), edible fresh mass (FM, g), and edible dry mass (DM, g) and are shown in Fig. 3). TLA was calculated with image recognition software, and manual corrections were performed as needed (see Supplementary Fig. 8). The edible portions of the plant include the leaf and shoot mass, but not the root mass. The results from the three film treatments across all three repeated experiments are summarized in Table 2.

A two-tailed student's $t$ test was performed on the model that included two factors, each at three levels which tested all pairwise comparisons of the effect least squares mean. These results provided statistically significant data that the QD films improved overall crop yield. The photosynthetic efficiency of the plants was improved because of the different light quality from the QD films, under which more biomass was produced per mol of photons than under the Control film. For vegetative growth in light-limiting environments, when DLI is reduced, the biomass accumulation is linearly reduced for most crops[33–38].

In these plant experiments, however, this degree of linearity was not observed for the R-QD treatment. Even with a reduction in DLI of 12% under the R-QD film, statistically significant increases were observed in edible DM, FM, and TLA. Thus, the light use efficiency (LUE, g mol$^{-1}$) was greater under the R-QD film, which directly correlates with the photosynthetic efficiency of the plant. The LUE was calculated by dividing the grams of fresh or dry mass produced per day by the DLI (see Supplementary Table 3). Table 2 illustrates how the R-QD film and O-QD films exhibited an improved LUE compared with the Control. In a recent study, it was shown that FR photons (700–750 nm) may be as photosynthetically efficient as PAR photons when they act synergistically with shorter wavelength photons[32]. If the photon flux density of this expanded waveband (400–750 nm) were calculated

for each of the three treatments, the results are 402, 423, and 378 µmol m$^{-2}$ s$^{-1}$ for the Control, O-QD, and R-QD treatments, respectively. This equates to a +5.2% increase for the O-QD film and only a −6% reduction under the R-QD film compared with the control. Interestingly, the increased intensity under the O-QD film for the new photosynthetically active radiation range (400–750 nm), as defined by ref. 32[32], correlated well with the LUE improvement for the O-QD treatment. Alternatively, the adjusted intensity values did not directly correlate with the LUE improvement for the R-QD, as a reduction in photon flux remained, even for the expanded waveband.

**QD films for space applications**. From the plant experiments above, it was demonstrated that modifying the solar spectrum with luminescent QD films would lead to increased plant production by improving the photosynthetic efficiency of the plant. If this technology were deployed in greenhouses on the Moon or Mars, additional benefits to plant production could be realized due to the larger availability of UV photons for conversion into PAR. To estimate these benefits, a simple model described below was employed.

First, the absorption spectrum of the O-QD and R-QD films (see Supplementary Figs. 18, 19) was used to calculate the percentage of absorbed photon flux for both terrestrial and extraterrestrial locations. This was performed by convoluting each QD film absorption spectra with the solar spectrum for reference air mass (AM) 1.5 on Earth and AM 1.0 in space (see Supplementary Figs. 22, 23 for the methodology)[39]. With the convoluted spectra, the absorption percentage of UV (<400 nm), B (400–500 nm), and G (500–600 nm) were calculated for each film by integrating the area under the convoluted absorption curve for each wavelength range (see Fig. 4).

After calculating how much light is absorbed for a given wavelength range (see Supplementary Table 5), the total photon flux density converted into PAR from the emission of the QDs was calculated. As a first approximation, the following assumptions were included: (1) the QY of the films was 100%, (2) the emission spectrum of the QDs was fully within the PAR region (400–700 nm), and (3) all emitted photons were directed toward the plants. Considering these assumptions, the net change in PAR would simply be the addition of the absorbed UV photon flux density that is converted into PL, since the converted blue and green photons to PL would remain within PAR under these assumptions. However, loss mechanisms present in a real system are unaccounted under these assumptions and therefore must be considered to obtain a more realistic estimate. The first loss

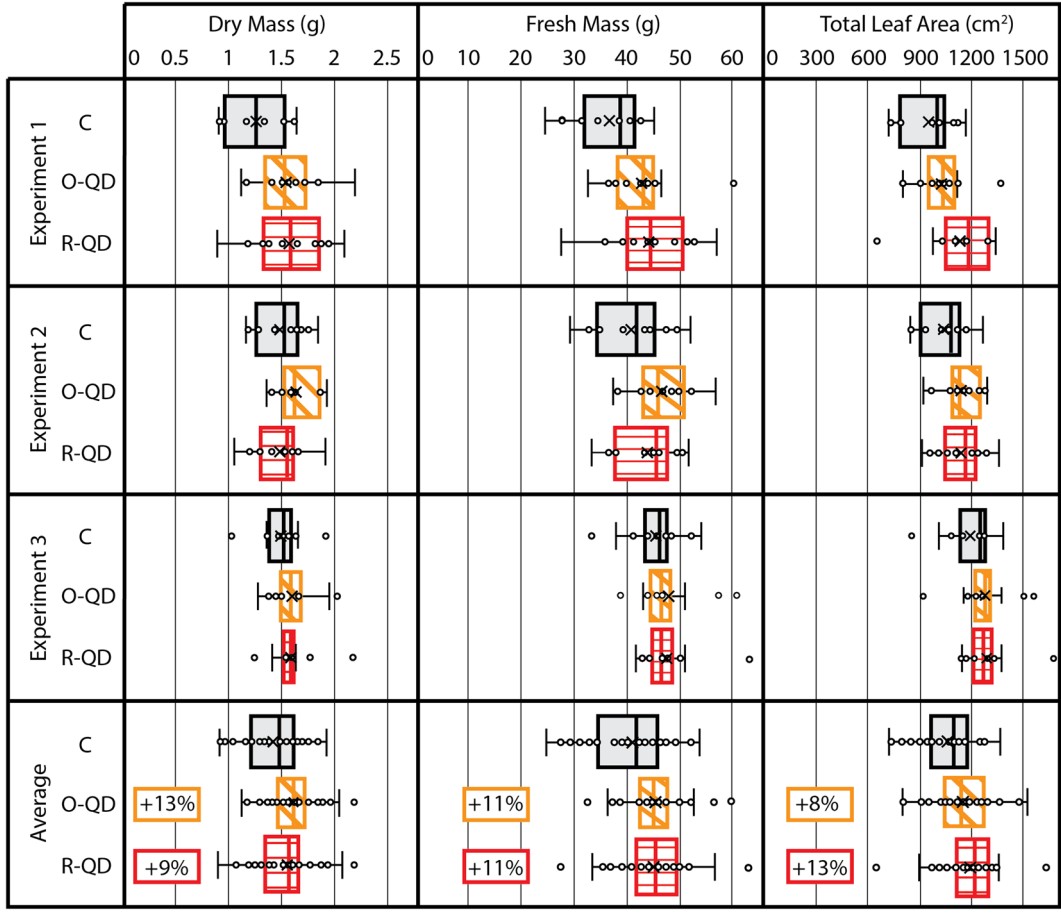

**Fig. 3 Experimental results of lettuce plant trials.** Experimental results of measured edible dry mass, edible fresh mass, and total leaf area for three replicate lettuce experiments under three different light spectra ($n = 12$ per group in each treatment; $N = 36$ per experiment). Error bars show one standard deviation. Cohen's $d$ effect sizes for each metric were statistically significant with a confidence level of 95%.

**Table 2 Percentage differences and significance of effect sizes, and calculated lighting-use efficiency (LUE) of O-QD and R-QD Films compared with Control (C).**

| | O-QD film (600 nm) | | | R-QD film (660 nm) | | |
|---|---|---|---|---|---|---|
| | DM | FM | TLA | DM | FM | TLA |
| Increase v. C (%) | +13% | +11% | +8.0% | +8.7% | +11% | +13% |
| $p$ value ($\alpha = 0.05$) | 0.003 | 0.004 | 0.02 | 0.05 | 0.005 | 0.0003 |
| LUE v. C (g mol$^{-1}$) | +5.6% | +3.9% | – | +17.6% | +17.2% | – |

Statistically significant increases were observed in dry mass (*DM*), fresh mass (*FM*), and total leaf area (*TLA*) at confidence level $\alpha = 0.05$.

mechanism to consider is the QY of the QD films not being 100%. In this case, both QD films had a measured PL QY of 85%. Second, due to the isotropic PL emission, it can be assumed that ~25% of the PL would not be absorbed by the plants due to PL emission in the opposite direction. Third, as mentioned previously, there are portions of the emission spectrum for both QD films that extended beyond the PAR region and thus wouldn't contribute to PAR. Including these three loss mechanisms, the estimated net difference in PPFD contributions were calculated for the O-QD and R-QD films under both the AM 1.0 and AM 1.5 solar spectra. Table 3 summarizes the results.

For the AM 1.0 spectrum, the PPFD below the O-QD film would in theory be increased by ~63 µmol m$^{-2}$ s$^{-1}$ (+2.6%), while the R-QD film would have an overall decrease in PPFD (−2.4%). For the AM 1.5 spectrum, neither of the films would be expected to improve PPFD, with respective decreases of −1.6% and −5% under the O-QD and R-QD films. Comparing the net ΔPPFD for the QD films under AM 1.0 and AM 1.5 solar spectra shows that there would be a relative improvement of +4.2% in PPFD for the O-QD film and +2.6% relative improvement in PPFD for the R-QD film under the AM 1.0 spectrum.

From these calculations, it can be concluded that for both solar spectra the net PPFD would be expected to decrease under the R-QD film, but a potential net increase in PPFD under the O-QD film would be expected for the AM 1.0 spectra. For the AM 1.5 spectra, a slight reduction in PPFD would still be expected for the O-QD film due to inherent losses in the system and to less UV light availability for shifting into PAR. For vegetative growth in light-limiting environments, biomass accumulation is positively correlated with DLI, generally in a linear trend[33], If we assume 1% DLI improves yield by 0.8% for lettuce[40], the estimated additional benefit for crop production in space applications would be 3.4% and 2.1% for the O-QD and R-QD films, respectively, due to converting more UV photons. This would be in addition to the observed increased production due to spectral quality shown in these plant experiments. For terrestrial applications, a net increase in PPFD would not be expected under either QD film; therefore, improved production would only be expected from spectral quality.

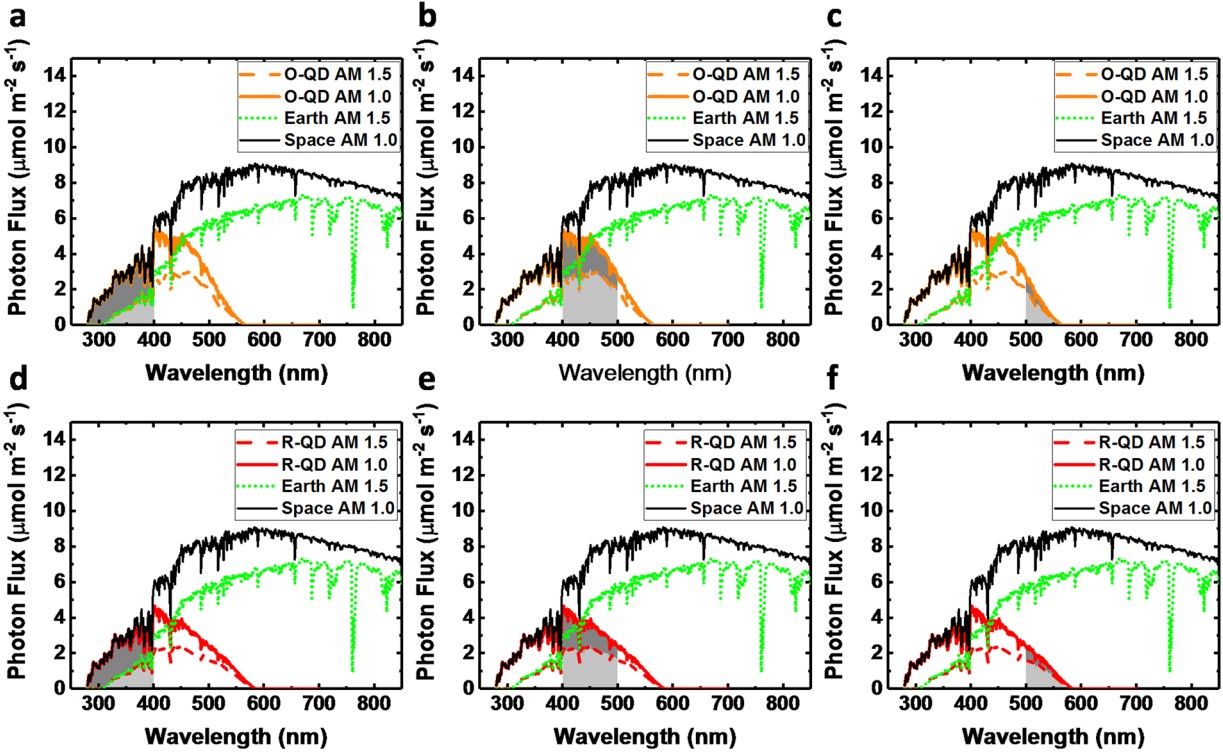

**Fig. 4 Comparison between absorbed photon flux for QD films under AM 1.0 and AM 1.5 spectra.** Convoluted absorption spectra of the O-QD film with the AM 1.0 and AM 1.5 spectra are presented in (**a**–**c**), where the integrated areas of absorbed photon flux of the (**a**) UV (<400 nm), (**b**) blue (400–500 nm), and (**c**) green (500–600 nm) spectral ranges are calculated for each solar spectra and compared. Similarly, convoluted absorption spectra of the R-QD film with the AM 1.0 and AM 1.5 spectra are presented in (**d**–**f**), where the integrated areas of absorbed photon flux for the (**d**) UV, (**e**) blue, and (**f**) green spectral ranges are calculated for each solar spectrum and compared.

**Table 3 Estimated changes in PPFD due to absorption and emission of QD films for AM 1.0 and AM 1.5 solar spectrum.**

| Film | Total PPFD ($\mu$mol m$^{-2}$ s$^{-1}$) | Net $\Delta$PPFD ($\mu$mol m$^{-2}$ s$^{-1}$) | $\Delta$PPFD UV (%) | $\Delta$PPFD blue (%) | $\Delta$PPFD green (%) | Net $\Delta$PPFD (%) |
|---|---|---|---|---|---|---|
| AM 1.0 | | | | | | |
| O-QD | 2413 | +62.6 | +8.6% | −5.1% | −0.9% | +2.6% |
| R-QD | 2413 | −58.3 | +6.2% | −6.6% | −2.0% | −2.4% |
| AM 1.5 | | | | | | |
| O-QD | 1735 | −26.8 | +3.7% | −4.3% | −0.9% | −1.6% |
| R-QD | 1735 | −86.0 | +2.6% | −5.6% | −2.0% | −5.0% |

It is noteworthy that the AM 1.0 spectrum used in these calculations was taken from the ASTM standard ASTME-490, which is based on irradiance measurements made by satellites and other equipment, and relates to the intensity at approximately the distance of Earth to the Sun[39]. In space missions, such as to the Moon or Mars, the intensity would be reduced by ~$1/d^2$, where $d$ is the distance from Earth to the new location and could be further reduced by a different atmosphere. Since the Moon has no atmosphere and it orbits the Earth, a reduction in available PPFD would not be expected. However, in a location like Mars, the intensity would be reduced to ~1/3 of the AM 1.0 spectrum measured at Earth, due to Mars being ~1.5× farther from the Sun and having an atmosphere, which further reduces the UV portion by an additional ~4%[41]. Since ~39% more PPFD is available under an AM 1.0 spectrum (measured at Earth) compared with the AM 1.5 spectrum, this would partially make up for the PAR reduction at Mars, but an overall reduction in PPFD compared with terrestrial applications would be expected. In spaceflight, crewed missions could integrate QDs into spacecraft crop production systems as remote phosphors for electrical lamps or

to modify optically waveguided light transmission from a connected solar concentrator[42–44]. These considerations further demonstrate the importance of maximizing the LUE for the plants, as the intensity of sunlight would continue to drop as crewed space missions expand farther out into the solar system.

In summary, this work introduced and demonstrated the benefits of a luminescent film technology enabled by CIS/ZnS QDs to improve spectral quality for plant growth. The edible DM under the O-QD and R-QD films was increased by +13% and +9%, respectively, and the edible FM was increased by +10% under both films. The TLA was also improved under the O-QD and R-QD films by +8% and +13%, respectively. These results clearly indicate that lettuce grown under the QD films exhibited more efficient growth than lettuce grown under the Control film, despite a DLI reduction of 12% under the R-QD film. This is supported by LUE calculations, where the DM was 18% greater under the R-QD film and 6% greater under the O-QD film. Finally, a mathematical model was implemented to estimate crop yield improvements under QD films when deployed in the AM 1.0 solar spectrum in space. The plant production improvements

from these QD films would potentially increase crop yields in CEA cultivation on Earth, and, due to their relative light weight and small form factor, would be beneficial for long-duration crewed space missions.

## Methods

**Plant growth test chamber design and operation**. Lettuce plant responses to the modified spectrum of light provided by the QD films were determined within a custom-built PGTC. The PGTC framework ($1.8 \times 1.0 \times 2.2$ m, L × W × H) consisted of three platforms (from top to bottom): a lamp mounting platform located 61 cm above the test films support platform, which was 91 cm above the plant growth platform.

A nutrient storage tank was located below the plant growth platform at the base of the PGTC. The lamp platform had four P.L. Light Systems luminaires fitted with MH lamps (SolisTek 400 W 10 K Finisher) to provide uniform radiation distribution above the test film support frame ($1.8$ m × 1 m), that held three horizontally mounted and adjacent test films. These 400-W luminaires were selected for the high light uniformity afforded by their parabolic reflector, and these MH lamps were selected as the high-intensity discharge lamps that best approximated the solar spectrum within the operational budget of this study. The plant growth platform was comprised of a $1.8$ m × 1 m root zone nutrient solution tray, shared by plants grown within each of three separate zones ($60 \times 86 \times 91$ cm, L × W × H). The zones were delineated by a white-on-black plastic film (California Grow Films ORCA Grow Film, 98% reflection and 99% diffusion of the visible spectrum) that was installed to maximize light intensity, ensure light uniformity, and separate each light treatment by eliminating cross-contamination of light between zones. The light passing through the films to the plant canopies below provided spatially uniform illumination of ~380 µmol m$^{-2}$ s$^{-1}$ PPFD and had a spectrum that was similar to the solar spectrum (see Supplementary Fig. 21).

Continuous monitoring was performed for other plant growth parameters, including the air and root zone temperatures, atmospheric $CO_2$ concentration, and nutrient solution pH and electrical conductivity (EC). These parameters remained equivalent for the plants within each of the three zones, thereby having only the light quality based on the light spectrum distribution as modified by the test film as a variable (see Supplementary Figs. 10–16 for characteristics of environmental uniformity). A ventilation fan was installed in each test zone to exchange cool, dry air from the room with the warm, moist air of the test zone and to maintain ambient $CO_2$ concentration, while providing air movement among the plant leaves.

The plants were secured and spaced 12 cm × 12 cm (55 plants m$^{-2}$, see Supplementary Fig. 17 for plant layout) within a rigid, high-density polyethylene board ($61 \times 100 \times 2.5$ cm, L × W × H) covered with ORCA Grow Film and set on top of a water-tight basin. A modified Hoagland hydroponic plant nutrient solution (see Supplementary Table 4 for formulation) was continually distributed and recirculated to the common root zone environment of all of the plants. The nutrient solution was maintained at a constant depth of 2.5 cm in the root zone tray by subirrigation from the nutrient solution storage tank (83 L), providing one volume change of solution every 5 min.

A data acquisition and control system (Campbell Scientific 21X Micrologger® datalogger) monitored and recorded 15 min averages of plant microclimate conditions of the PGTC measured at 5 s intervals. The datalogger also activated the lamps to control the photoperiod (12:00–02:00, 14 h), and it controlled the nutrient solution pH (6.1–6.2; Weiss Research PHS-0201-3B). Quantum sensors (400–700 nm; Apogee Instruments SQ-500-SS; error ± 0.8%) and radiation-shielded thermistors (Campbell Scientific 107 BetaTherm 100K6A1IA, error ± 0.5 °C) were installed in each test zone of the PGTC to measure PPFD (µmol m$^{-2}$ s$^{-1}$) and air temperature (°C), respectively. Average photoperiod/scotoperiod air temperatures were 24/20 °C, respectively, and average air temperature differences between each test zone remained within ±0.5 °C throughout the experiments. The atmospheric $CO_2$ concentration (ppm; Vaisala GMT222, error ± 1.5%) was monitored in the room adjacent to the PGTC but not controlled and ranged from 361376 ppm for each experiment.

The recirculating, deep-water culture hydroponic nutrient delivery system consisted of a pump, four emitters, one drain, and two cylindrical storage reservoirs connected with each other at both ends. The shared nutrient solution temperature (°C, Type T copper-constantan thermocouple, error ± 1.0 °C) and electrical conductivity (EC, mS cm$^{-1}$; Hanna Instruments HI-3001) were measured and recorded. The acidity of the nutrient solution was controlled to maintain a pH setpoint (6.1–6.2) by the addition of dilute nitric acid into the nutrient storage reservoir by a peristaltic pump. The nutrient storage tank was monitored and manually refreshed daily to maintain EC setpoint (1.8–1.9 mS cm$^{-1}$) by addition of premixed nutrient solution. Nutrient solution and/or tap H2O was manually added to the storage reservoirs on a daily basis to ensure the EC remained on target. The nutrient solution was pumped up from the reservoirs to the back wall of the PGTC, where the manifold of four equidistant emitters constantly output nutrient solution to the lettuce plants in the shared basin. The nutrient solution drained from a single drain at the opposite end of the basin, where it returned to the nutrient storage reservoirs for recirculation.

**Plant studies in the PGTC**. In each plant experiment, lettuce plants were established by double-seeding 150, 3.8 cm rockwool starter cubes (Grodan AO 36/40)

with red romaine lettuce (*Lactuca sativa* L. cv. "Outredgeous"; Johnny's Seeds, organic variety). They were placed in the three plant zones in the PGTC for 48–72 h without light till seed germination was observed. Germination was marked by root radicle emergence from the seed, after which a 14 h photoperiod (14 h of light per 24 h period) was initiated to provide a target average daily light integral (DLI) of 17 mol m$^{-2}$ d$^{-1}$ (see Supplementary Table 2 and Supplementary Figs. 5–7 for DLI values). At 7 DAS, the first true leaves emerged, and thinning was performed to leave one healthy seedling in each rockwool cube. Extra rockwool cubes were removed from the floating raft within each of the three treatment zones to achieve a final planting density of 55 plants m$^{-2}$. The experimental treatments continued to harvest at 28 DAS.

The rockwool cubes were irrigated continuously with pH 6.2 tap water until germination, and between germination and thinning they were irrigated with half-strength modified Hoagland's nutrient solution. After thinning, full-strength solution was used in the recirculating hydroponic basin which continuously pumped stored nutrient solution to plants in all three zones for the duration of the experiment at a system turnover rate of 5 min.

Harvest occurred at 28 DAS when the edible fresh mass (g) was measured for each data plant ($n = 12$ per treatment, $N = 36$ total). Leaves were then flattened and photographed for measurement of total leaf area (TLA, cm$^2$) with image analysis software (see Supplementary Information for software description). After photographing, each plant sample was individually bagged and labeled, dried in a drying oven at 40 °C for 72 h to ensure thoroughly uniform drying (see Supplementary Fig. 9 for percentage dry mass values), and weighed to determine dry mass (g).

Between experiments, spectra under the O-QD and R-QD films were measured with a spectroradiometer (Apogee Instruments, PS-300) and compared with the spectra under the C film.

**Statistics and reproducibility**. A two-tailed student's $t$ test was performed on the model that included two factors, each at three levels which tested all pairwise comparisons of the effect least squares mean. The levels included O-QD, R-QD, and Control (C) film treatments, while the second factor included replicate experiment numbers 1, 2, and 3. Cohen's $d$ effect sizes were calculated to determine the percentage difference in edible DM, edible FM, and TLA between each QD film and C. The effect size gauged the contribution of only one factor among multiple factors (film treatment and experiment number), and as such showed the significance of the contribution from the film treatment only, despite noise in the variation from combining both factors.

**QD film manufacture and optical properties**. The CuInS$_2$ (CIS)/ZnS QDs used in this study were manufactured using methods similar to those described in reference[45] with slight modifications to optimize peak wavelength emission and QY. Size of the orange (590 nm PL emission) and red (630 nm PL emission) QDs are $4.8 \pm 0.7$ and $5.1 \pm 0.9$ nm, respectively, and have typical size distribution of ~15–19%. The CIS QDs were incorporated into agriculture films 60 cm × 91 cm in size by dispersing the nanoparticles into an acrylic resin (described in reference[46]), which was then coated between two sheets of polyethylene terephthalate (PET) barrier film (manufactured by I-Components, Co.) using a drawdown method. The water vapor transmission rate for the PET barrier films was 0.09 (g/m$^2$/day). The laminate film was then exposed to 405 nm-emitting LEDs to cure the nanocomposite interlayer. The total thickness of the films was 350 µm with an interlayer thickness of 150 µm. After curing the QD resin between the two PET layers, the optical properties of both films were characterized. The PL QY was measured with a commercial PL spectrometer (Horiba Fluoromax 4) equipped with an integrating sphere and using an excitation wavelength of 440 nm. The QY of each film was measured to be 85% ± 5%, with a peak emission for the O-QD film at 600 nm (120 nm FWHM) and at 660 nm (120 nm FWHM) for the R-QD film. The haze of both QD films and the bilayer, 6 mil polyethylene (PE) C film was measured using a custom-built setup consisting of a collimated 640 nm LED light source, a 30.5 cm diameter integrating sphere, and a fiber optic spectrometer (Avantes AvaSpec Dual Channel). The haze for all three films was measured to be 2% ± 0.5%.

**Reporting summary**. Further information on research design is available in the Nature Research Reporting Summary linked to this article.

## Data availability

The data that support the findings of this study are available from the authors on reasonable request, see author contributions for specific data sets.

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

## Acknowledgements

This work was funded under a Small Business Technology Transfer contract by the National Aeronautics and Space Administration under Award No. 80NSSC18P2144 and the University of Arizona ALVSCE Bridge Funding Program 2019. The total leaf area analysis program was developed and implemented by KC Shasteen, and consultation on statistical methods was provided by Lingling An, Ph.D. All experimental procedures with the PGTC were completed in the MLGH Laboratory at the University of Arizona Controlled Environment Agriculture Center.

## Author contributions

The experimental setup and scope of the project were the result of interactions between M.R.B., G.A.G., and H.M. QDs were optimized and manufactured by K.R. The QD films were optimized and manufactured by A.J. The experiments were planned by M.R.B., G.A.G., C.H.P., and H.M. The spectroscopic characterization of the QD films was performed by M.R.B., A.J., and D.H. Modeling was conducted by M.R.B. The design, operations, and plant trials in the PGTC to obtain the spectroradiometer measurements were completed by C.H.P. and G.A.G. Data was analyzed by C.H.P., G.A.G., M.R.B., and D.H. The paper was written by C.H.P., G.A.G., and M.R.B., in consultation with all the authors.

## Competing interests

Authors C.H.P., D.H., A.J., K.R., H.M., and M.R.B. were employed by UbiQD, Inc. when the paper was submitted and may also have equity in the company. Authors C.H.P. and G.A.G. were not employed by UbiQD, Inc. and had no equity in the company while

conducting these experiments, but C.H.P. is currently employed by UbiQD, Inc. at the date of paper submission.
