## [Peer Review File · Communications Biology]

Reviewers' comments:

Reviewer #1 (Remarks to the Author):

Overall I am very pleased with the quality of this paper.

I only have a few comments.

1- Based on the results of the plant response I would not agree that you had a statistical difference between the treatments. It appears you went looking for a statistical method to allow for significance, where a student t or ANOVA would not provide you with the statistics you were looking for. I would suggest you list both styles to allow the reader to make the conclusions.

2- Likewise, I would like to see a statistical analysis of the wavelength shift to show that it was also statistically different not just the plant response.

3- I am never a big fan of the 1% for 1% change, this only occurs at low light levels and there are a number of other papers that show 2.5% light change impacts 1% of yield, I would suggest you cite both.

4- supplemental figure 9 - you need to provide a legend for the data, I do not know what bars belong to which treatments.

Reviewer #2 (Remarks to the Author):

This study has demonstrated the modification of an incoming solar spectrum through downconversion of high energy UV/blue photons into orange/red photons using luminescent quantum dot films, leading to improved growth of red romaine lettuce. I find the concept and approach of this work to be incredibly innovative and it would provoke considerable interest to a vast audience in the areas of both terrestrial and space-based agriculture. The title is suitable for the content, and I believe this work would be a significant contribution, but there are major elements I have immediate concerns about below that need to be addressed.

1. If I understand correctly, the claim is made that the PPFD under the R-QD film exhibited a reduction due to emission extending into the far red that the PAR sensor did not measure. But were there attempts to mitigate this reduction by boosting the intensity of the metal-halide lamps above the R-QD film only, to make the PPFD under the film comparable with the O-QD and Control? Perhaps by slightly lowering one of the lamps or installing an additional lamp on that side? I understand this would concomitantly boost the FR along with PPFD, but increasing incident photons should not alter the ratios, and the impact on growth would inherently be the effect of the R-QD film.

2. On page 6, the manuscript states "spectra under the R-QD and O-QD films were measured with a spectroradiometer (Apogee PS-300)," but the spectral distributions presented in Supplemental Figure 20 state the PPFD spectra underneath the films were measured with an SRI-PL-6000 Spectrophotometer. Please clarify.

3. The text under Figure 2 states spectral quality was measured from 300 to 850 nm, but the wavelength range in the graph stops at 800 nm. Please address.

4. I think Supplemental Figure 20 is scientifically more vital to display in the manuscript as Figure 2, than the current "Changes in spectral differences of QD-films over time" Figure 2. No doubt, it is important to present the stability of the films throughout each replication, but I think this could be mentioned as you have it, and the readers referred to it as a Supplemental Figure. On the other hand, I find the PPFD spectra (Supp. Figure 20) to be a more compelling feature for the purpose of the QD films. The spectra not only characterize the crop lighting environment under the films, but they graphically illustrate and confirm what is quantified in Table 1, the absorption of the UV/Blue

wavelengths relative to the control, and the enhanced Orange/Red emissions contributed by the QD films, appealing to both numerically and visually adept readers. This spectral modification is what ultimately led to the improved plant growth, and it should be accentuated and easily viewable during reading by being placed inside the body of the manuscript.

5. If an Apogee PS-300 was used, the software should allow the user to determine the measurement range all the way out to 1100 nm. If the PAR sensor did not measure beyond 700 nm, how was the FR intensity measured and %'s determined? Even if ratios were determined solely by calculating the integral under the spectra, the FR and UV intensities should have still been measured separately with the spectroradiometer to confirm, by setting the measurement ranges to 700–800 nm, and 350–400 nm for FR and UV, respectively. For UV, I didn't see much before 350 nm.

6. In regard to your claim that "the light intensity for each treatment area were equal," I am assuming this refers to the area above the QD films? I am concerned that the difference in PPFD under the R-QD treatment is quite considerable, and introduces another variable (intensity) that directly impacts biomass accumulation, in addition to the spectrum variable.

7. I am concerned that your PPFD values as presented in Table 1, are not presented correctly. You have UV and FR as %'s of the total PPFD, when they should be presented separately from PPFD ratios as simply $\mu\text{mol}/\text{m}^2/\text{sec}$. If the presented PPFD values include contributions from UV and FR, then your actual total PPFD values (400-700 nm) are considerably lower than what you targeted for the experiment. If UV and FR are excluded from your total PPFD, then your actual PPFD values as presented in Table 1 are 335 (16.8 DLI), 341 (17.1 DLI), and 276 (13.9 DLI) $\mu\text{mol}/\text{m}^2/\text{sec}$ for the Control, O-QD, and R-QD treatments, respectively. This means, the true PPFD difference under the R-QD film is actually 18 and 19% lower than the Control and O-QD treatments, respectively. In considering your statement, "a 1% reduction in DLI correlates with a 1% reduction in biomass production for most plants," this PPFD difference under the R-QD film is too large for a fair comparison of spectral effect, and your current results may be dramatically different if the PPFD under the R-QD film had been at least within 3% of the Control PPFD. Regrettably, I have to suggest that the growth experiment be repeated, ensuring the PPFDs under both film treatments are comparable and both within at least 3% of the Control. Or, either remove the results of the R-QD film from the manuscript.

8. For Table 1, I recommend that you present the PPFD values of each radiation category in $\mu\text{mol}/\text{m}^2/\text{sec}$, and only the RGB ratios as %'s in parentheses next to their PPFD value. You may also want to consider inserting a column for R:FR ratio (to one decimal place).

9. Just curious, with such a large Stokes shift, would increasing the intensity of the light source above, decrease the PL QY of the R-QD film? Perhaps through increased non-radiative dissipation (thermal relaxation)?

10. Would using an LED-based artificial sunlight fixtures as the light source provide you with more control over the intensity and performance of the QD films (e.g. dimming capability, even closer resemblance to solar spectrum, single fixture for each treatment...etc)?

From this point onward, I will comment and provide recommendations on specific sections heading by heading to aid the authors in improving the flow and format of the manuscript.

11. Abstract:

a. Lines 3-5: Consider revising to say, "CuInS₂/ZnS quantum dot (QD) films were used to down-convert ultraviolet/blue photons to red emissions at 600 and 660 nm resulting in increased biomass accumulation in red romaine lettuce." Reads better.

b. Line 5 (second sentence): Revise to say, "All plant growth parameters except for spectral

quality, were held constant..."

c. Line 10: Revise to say, "enhance productivity in greenhouses on Earth, or in space where further improvements are expected..."

12. Introduction:

a. You are missing an Introduction heading at the top of page 2.

b. 2nd paragraph, line 2: After "crewed space applications," consider inserting the citations from the Zabel et al. 2016 and Wheeler, 2017 references below:

Zabel, P., Bamsey, M., Schubert D., & Tajmar, M. Review and analysis of over 40 years of space plant growth systems. *Life Sciences in Space Research*. 10, 1–16 (2016).

Wheeler, R.M. Agriculture for Space: People and Places Paving the Way. *Open Agriculture*. 2, 14-22 (2017).

c. 4th paragraph, line 1: The use of terms such as "new", "novel", or "for the first time" should be avoided. I suggest replacing "A new technology" with "Here, we present an approach" that uses luminescent...etc.

The novelty is in your work, and it speaks for itself.

13. Results:

a. I recommend this section be designated as a "Results and Discussion" section. I find the sections entitled, "Spectral Quality on Plant Growth," and "Spectral Control with CIS/ZnS QD Films" to be highly introductory in nature and contain a lot of background, and not appropriate to be presented in a Results section.

b. I recommend the entire "Spectral Quality on Plant Growth" section as is (remove subheading title), be moved and inserted after the 2nd paragraph of the Introduction, and remove the reference to Figure 1a (photosynthetic action spectrum) in the second paragraph. The citations of McCree's papers are sufficient here since you refer to this figure again later on (page 5).

c. In the 1st paragraph of "Spectral Quality on Plant Growth," lines 4-6, this sentence should be rearranged. I understand what you are trying to communicate, but the presentation of these fundamental concepts seems out of order. I suggest revising to the following, "The PPFD a plant receives in a 24-hour day is measured as the daily light integral (DLI, mol m⁻² d⁻¹), and is directly proportional to the rate of plant growth which is subsequently determined by biomass accumulation."

d. I recommend the 1st paragraph of the "Spectral Control with CIS/ZnS QD Films" section be moved and inserted after the 3rd paragraph of the Introduction. This paragraph presents a solid background on QD Films, but it is in the wrong place. I believe if you insert it after discussing the gaps/challenges to controlling light quality, it would serve as an effective connecting paragraph to the final Introduction paragraph that summarizes your approach and what you did.

e. Consider starting the Results and Discussion with the 2nd paragraph of the "Spectral Control with CIS/ZnS QD Films" section. However, the entire paragraph 4 and the first sentence of paragraph 5 should be moved to the Methods section.

f. All statistical analyses appear to be appropriate for the experimental design, but the details should probably be presented in the Methods.

g. There is an additional aspect you could mention in the "QD Films Applications for Space" section. Your approach could be also employed during space transit as well. Either en route to

Mars, in cis-lunar space, or even aboard the ISS. You may want to mention the potential application of your QD films as coatings for lighting arrays that could be connected to solar concentrators via optical waveguides that could transmit solar light directly into growth chambers, and modifying the spectrum as the final step. For reference, see Nakamura et al. 2010 and Nakamura et al 2012 below.

Nakamura, T., Van Pelt, A.D., Yorio, N.C., Drysdale, A.E., Wheeler, R.M., & Sager, J.C. Transmission and Distribution of Photosynthetically Active Radiation (PAR) from Solar and Electric Light Sources. *Habitation*, 12, 103-117 (2010).

Nakamura, T., & Smith, B.K. Space Solar Energy System for Thermal and Photosynthetic Applications. *American Institute of Aeronautics and Astronautics Space*, 5168, 1-13 (2012).

Also see report by Mickens et al. 2019 below that mentions potential use of harvesting sunlight directly for growth chambers on space modules during transit. This group also used an artificial sunlight fixture comprised entirely of LEDs that could serve as alternative light source above the QD films of your experiment.

Mickens, M.A., Torralba, M., Robinson, S.A., Spencer, L.E., Romeyn, M.W., Massa, G.D., & Wheeler, R.M. Growth of red pak choi under red and blue, supplemented white, and artificial sunlight provided by LEDs. *Scientia Horticulturae*, 245, 200-209 (2019).

14. Discussion:

This section should not be designated as a stand-alone Discussion. A discussion consists of an extended analysis of the results and their comparison to the literature rather than a short summary or conclusion. Although this paragraph is well-written and suitable as a summary, I suggest making it the final paragraph of the previous section, "QD Films for Space Applications" for closure.

15. Methods:

All aspects of the methodology are appropriate with sufficient detail for replication. However, I do suggest a "Statistics and Reproducibility" section be inserted somewhere in this section. A lot of statistical analysis details were presented back in the Results section entitled, "Spectral effects on red romaine lettuce production," and perhaps should be moved here.

Reviewer #3 (Remarks to the Author):

This paper reports on the enhancement of the greenhouse crop production which has been obtained by using a coverage made by a polymeric film doped with luminescent quantum dots (QDs). These latter absorb a fraction of the solar UV spectrum and re-emit light in the red spectral region which is especially beneficial for the growth of many vegetables. The main claim of the paper is that:

"no viable options were available for controlling the quality of light for greenhouse ...".

However, this claim is completely false. Indeed, there are plenty of examples of films doped with different dyes which are able to provide this same enhancement. Just as an example you can read this paper (the very first one I found in the web) *Europ. J. Hort. Sci.*, 71 (4). S. 145-154, 2006 and the therein references. Furthermore, I am pretty sure that in the 1990s films doped with Europium complexes had even been marketed for this same purpose.

The reported results do not show any peculiar advantage in the use of QDs instead of conventional

organic dyes for the light conversion. So, I do not find in the paper any element of novelty that supports the publication of this work in high impact factor journals.

Response to Reviewer's comments below in bold.

Reviewers' comments:

Reviewer #1 (Remarks to the Author):

Overall I am very pleased with the quality of this paper.

I only have a few comments.

1- Based on the results of the plant response I would not agree that you had a statistical difference between the treatments. It appears you went looking for a statistical method to allow for significance, where a student t or ANOVA would not provide you with the statistics you were looking for. I would suggest you list both styles to allow the reader to make the conclusions.

We thank the reviewer for noting the importance of statistical significance. Our statistical analysis was indeed a student's t-test. The raw data were analyzed after the removal of outliers (> 3 standard deviations from the mean), which is standard practice. Only one outlier out of N=108 was removed based on this criterion. We had, however, included an artifact in the original paragraph describing the statistical analysis from previous work denoting that logarithmic transformation had been performed. This transformation was not performed on the data presented here. This inaccurate note has now been removed, and additional clarification has been added to this section and to the methods section describing the statistical analysis.

2- Likewise, I would like to see a statistical analysis of the wavelength shift to show that it was also statistical different not just the plant response.

To show statistical significance, we direct the reviewer to supplemental Figure 20 (originally Figure 2 in the manuscript). Here we have subtracted the spectrum under the O-QD film from the control spectrum for each of the three experiment (a) and similarly we did this subtraction for the R-QD film from the control for each experiment (b).

In these plots, a comparison is made on how the spectrum differs from measurements made under the two QD film light treatments compared to measurements taken under the control. Comparing the local minima (i.e. absorption) and maxima (i.e. emission) to the noise for both cases, we see they differ by about an order of magnitude. We also see that the Δ PPFD difference for the O-QD and R-QD films compared to the control are 1-2 $\mu\text{mol m}^{-2} \text{s}^{-1}$ where the overall spectra (Figure 2) show a signal $\sim 5 \mu\text{mol m}^{-2} \text{s}^{-1}$ (excluding the two spikes between 550-600 nm). Therefore, we observed a spectral shift on the order of $\sim 20\%$, which has practical significance.

3- I am never a big fan of the 1% for 1% change, this only occurs at low light levels and there are a number of other papers that show 2.5% light change impacts 1% of yield, I would suggest you cite both.

The 1% for 1% change rule-of-thumb, often referenced, can indeed be too simplistic, as it depends on environmental conditions as well as crop variety. It is safe to say a linear relationship between light intensity (PPFD) and biomass accumulation has been demonstrated for vegetative growth at relatively low light intensities with light being the limiting factor (see added references on p.9 of the manuscript). Other environmental parameters, such as CO_2 concentration and temperature can

influence and limit this trend, but at the light levels used in this study, a linear relationship between PPFD and assimilated CO₂ was expected per Taiz & Zeiger, 2010. Additionally, in the modeling section, we have also modified the assumption of estimated biomass accumulation of 0.8% for an increase in DLI of 1% reported for lettuce by De Pinheiro Henriques (2000). This is only used for an estimation of the potential added benefit for crop production in space due to a higher flux of UV photons.

4- supplemental figure 9 - you need to provide a legend for the data, I do not know what bars belong to which treatments.

Legend added.

Reviewer #2 (Remarks to the Author):

This study has demonstrated the modification of an incoming solar spectrum through downconversion of high energy UV/blue photons into orange/red photons using luminescent quantum dot films, leading to improved growth of red romaine lettuce. I find the concept and approach of this work to be incredibly innovative and it would provoke considerable interest to a vast audience in the areas of both terrestrial and space-based agriculture. The title is suitable for the content, and I believe this work would be a significant contribution, but there are major elements I have immediate concerns about below that need to be addressed.

1. If I understand correctly, the claim is made that the PPFD under the R-QD film exhibited a reduction due to emission extending into the far red that the PAR sensor did not measure. But were there attempts to mitigate this reduction by boosting the intensity of the metal-halide lamps above the R-QD film only, to make the PPFD under the film comparable with the O-QD and Control? Perhaps by slightly lowering one of the lamps or installing an additional lamp on that side? I understand this would concomitantly boost the FR along with PPFD, but increasing incident photons should not alter the ratios, and the impact on growth would inherently be the effect of the R-QD film.

We appreciate the reviewer's enthusiasm for this topic, and careful attention to the control condition. For this experiment, the arrangement of the lights was optimized to provide a consistent incident light intensity (*i.e.*, PPFD) over all three light treatments with the highest light uniformity that could be achieved in the area of the experiment (see Supplemental Figure 1). This would most closely mimic how the QD-films would behave in a greenhouse setting, but in this experiment, we are able to better control all other variables.

Additionally, since there is an open area between the lights and the test films, any additional light or adjustments to the lights would affect both the red treatment and the neighboring control treatment.

Arguably, if we were to increase the PPFD above the R-QD, then the R-QD would have a greater advantage over the other 2 treatments due to higher light intensity, and the experiment would be lost, because we know that more PPFD = more biomass accumulation (at least to a limit, assuming no other environmental deficiencies).

Far-red photons have also been shown to have synergistic effects with PAR photons, even acting to improve biomass accumulation in similar efficiencies as PAR itself when supplied in conjunction.

Refer, for example to Zhen & Bugbee (2020), which was published since the work herein was completed:

- Zhen, S., & Bugbee, B. Far-red photons have equivalent efficiency to traditional photosynthetic photons: Implications for redefining photosynthetically active radiation. *Plant Cell Environ.* 43, 1259-1272 (2020).

Additionally, by observing a biomass increase under the red treatment even though the lettuce received less PPFD, further illustrates the spectral effect on the plant response. By overcoming the lower light intensity and still showing a yield increase, the red treatment has exhibited the importance of spectral quality on the plant development. By performing the experiment in this manner, those results are still valid and the outcome of the experiment is more applicable to using the QD-films in a terrestrial greenhouse or on space missions.

2. On page 6, the manuscript states “spectra under the R-QD and O-QD films were measured with a spectroradiometer (Apogee PS-300),” but the spectral distributions presented in Supplemental Figure 20 state the PPFD spectra underneath the films were measured with an SRI-PL-6000 Spectrophotometer. Please clarify.

Clarification was added to designate between the spectra measured at the time of film manufacture (where the SRI-PL-6000 spectrophotometer was used to characterize the spectra) and the spectra measured between plant experiments in the lab, which used the Apogee spectroradiometer.

3. The text under Figure 2 states spectral quality was measured from 300 to 850 nm, but the wavelength range in the graph stops at 800 nm. Please address.

The plots have been corrected to show the entire 300-850 nm spectra (see Figure 2 and Supplemental Figure 20).

4. I think Supplemental Figure 20 is scientifically more vital to display in the manuscript as Figure 2, than the current “Changes in spectral differences of QD-films over time” Figure 2. No doubt, it is important to present the stability of the films throughout each replication, but I think this could be mentioned as you have it, and the readers referred to it as a Supplemental Figure. On the other hand, I find the PPFD spectra (Supp. Figure 20) to be a more compelling feature for the purpose of the QD films. The spectra not only characterize the crop lighting environment under the films, but they graphically illustrate and confirm what is quantified in Table 1, the absorption of the UV/Blue wavelengths relative to the control, and the enhanced Orange/Red emissions contributed by the QD films, appealing to both numerically and visually adept readers. This spectral modification is what ultimately led to the improved plant growth, and it should be accentuated and easily viewable during reading by being placed inside the body of the manuscript.

We have swapped the locations Figure 2 of the manuscript and Supplemental Figure 20 as recommended.

5. If an Apogee PS-300 was used, the software should allow the user to determine the measurement range all the way out to 1100 nm. If the PAR sensor did not measure beyond 700 nm, how was the FR intensity measured and %'s determined? Even if ratios were determined solely by calculating the integral under the

spectra, the FR and UV intensities should have still been measured separately with the spectroradiometer to confirm, by setting the measurement ranges to 700–800 nm, and 350–400 nm for FR and UV, respectively. For UV, I didn't see much before 350 nm.

In between each experiment, spectra were measured with the Apogee Instruments PS-300 spectroradiometer and accompanying StellarNet SpectraWiz software and indeed have a spectral range extending to 1100 nm (we show data only out to 850nm to ensure the reader can focus on UV (<400 nm), PAR (400-700 nm), and FR photons (700-800 nm). These measurements yielded FR and UV intensities that were used to determine percentages and ratios. Measurements were performed across the entire spectral range of the instrument. Indeed, the lamp spectral output had negligible intensity below 350 nm.

6. In regard to your claim that “the light intensity for each treatment area were equal,” I am assuming this refers to the area above the QD films? I am concerned that the difference in PPFD under the R-QD treatment is quite considerable, and introduces another variable (intensity) that directly impacts biomass accumulation, in addition to the spectrum variable.

Correct, the light intensity above each film was equal. Please refer to our response to comment #1 regarding the R-QD treatment and why even though the PPFD was lower, the plant response still showed an increase in biomass accumulation, further supporting the fact that the spectral quality had a large impact on the development of the crop.

7. I am concerned that your PPFD values as presented in Table 1, are not presented correctly. You have UV and FR as %'s of the total PPFD, when they should be presented separately from PPFD ratios as simply $\mu\text{mol}/\text{m}^2/\text{sec}$. If the presented PPFD values include contributions from UV and FR, then your actual total PPFD values (400-700 nm) are considerably lower than what you targeted for the experiment. If UV and FR are excluded from your total PPFD, then your actual PPFD values as presented in Table 1 are 335 (16.8 DLI), 341 (17.1 DLI), and 276 (13.9 DLI) $\mu\text{mol}/\text{m}^2/\text{sec}$ for the Control, O-QD, and R-QD treatments, respectively. This means, the true PPFD difference under the R-QD film is actually 18 and 19% lower than the Control and O-QD treatments, respectively. In considering your statement, "a 1% reduction in DLI correlates with a 1% reduction in biomass production for most plants," this PPFD difference under the R-QD film is too large for a fair comparison of spectral effect, and your current results may be dramatically different if the PPFD under the R-QD film had been at least within 3% of the Control PPFD. Regrettably, I have to suggest that the growth experiment be repeated, ensuring the PPFDs under both film treatments are comparable and both within at least 3% of the Control. Or, either remove the results of the R-QD film from the manuscript.

We thank the reviewer for raising this issue of PFD vs PPFD, we could have been more explicit about the spectral ranges we used. The percentages originally shown in Table 1 are related to the total spectrum (PFD, 300-800 nm) and the table has been modified to clarify. Regarding the FR and UV portions, the %s reflect the portion of the total spectrum as indicated in the modified Table 1 headings: 300-800 nm. These are not percentages of PPFD, as only the BGR wavebands are in the PAR range. The PPFD under the R-QD film was 12% lower than PPFD under the Control, which was 3% lower than under the O-QD film. Clarity has been added to the Table 1 description regarding the percentage interpretation.

Regarding the request to repeat the growth experiments “ensuring PPFd’s under both film treatments are comparable and both within 3% of the control”, we respectfully submit that the results from the current set of experiments should be considered complete as is. The reasoning for not increasing the light intensity over the red QD-film, as suggested, is three-fold.

1) The purpose of this study was to compare the spectral affects attributed to the photoconversion from QD-films, and so the most accurate setup to do this was to ensure the incident light was equal above each of the treatments. This most closely resembles the real-world application of sunlight as the light source and the QD-films altering the spectrum in a greenhouse on earth or in space.

2) Due to the setup of the grow chamber where the lights are not isolated for each treatment area (i.e. there is an open interstitial space between the luminaires and films) if we were to lower a fixture or include an additional fixture over the R-QD to increase the PPFd below the film, this additional light would also affect the intensity of the neighboring test area (the control), which then would affect the intensity differences between the control and O-QD film.

3) After initially submitting the manuscript, a new study was published (Zhen & Bugbee, 2020) showing that far-red photons can have similar photosynthetic efficiencies as PAR photons (400-700 nm) when sufficient PAR photons are present (they act synergistically), which we understand to be the case in our experiment. If these findings are correct, then by increasing the intensity over the R-QD film to ensure the PPFd (400-700 nm) to within 3% of the Control and O-QD treatments, the total amount of photons contributing to photosynthesis (400-800 nm) would be considerably higher for the red treatment.

The improved biomass accumulation under the R-QD film compared with the control, even with a lower PPFd, runs counter to the initially expected result of lower PPFd inducing lower biomass accumulation. However, due to the modified spectral quality, the lettuce grown under the R-QD films exhibited a statistically significant improvement in biomass accumulation and leaf area. These findings further support the conclusions made in the paper that the QD-modified spectrum improves photosynthetic efficiency in lettuce and could enhance productivity in greenhouses on Earth and in space. Therefore, the R-QD results comprise a meaningful component of this work, and we argue they should remain in the manuscript.

8. For Table 1, I recommend that you present the PPFd values of each radiation category in $\mu\text{mol}/\text{m}^2/\text{sec}$, and only the RGB ratios as %’s in parentheses next to their PPFd value. You may also want to consider inserting a column for R:FR ratio (to one decimal place).

Table 1 has been modified to include the total photon flux density over the spectral range 300-800 nm, with units of $\mu\text{mol m}^{-2} \text{s}^{-1}$. We have also included the %s in parentheses as suggested, where the %s are of the total PFD for each spectral waveband. We have also included R:FR as recommended.

9. Just curious, with such a large Stokes shift, would increasing the intensity of the light source above, decrease the PL QY of the R-QD film? Perhaps through increased non-radiative dissipation (thermal relaxation)?

We thank the reviewer for the question. We have not observed any changes in QY with light intensity in this excitation range. Increasing light intensity wouldn’t directly impact the QY of the R-QD film because we operate in a regime where QDs can only be excited by a single photon (re-excitation is slower than relaxation), however, the additional heat provided to the film by a higher

light intensity could lower the conversion efficiency of the QDs through increased non-radiative recombination (multiple exciton regime).

10. Would using an LED-based artificial sunlight fixtures as the light source provide you with more control over the intensity and performance of the QD films (e.g. dimming capability, even closer resemblance to solar spectrum, single fixture for each treatment...etc)?

While many reported spectra for broad-spectrum LEDs display interpolated values between actual peak wavelength emissions, phosphors have been incorporated in some models to broaden the emission across certain wavebands. Indeed, there are R&D lamps that show a reasonable spectral match with the sun, but these lamps were prohibitively expensive for this work and were not commercially available as far as the authors are aware. The lamps used herein exhibited a broad, relatively continuous emission across the waveband of interest and provided a sufficient spectrum that mimics the sun.

From this point onward, I will comment and provide recommendations on specific sections heading by heading to aid the authors in improving the flow and format of the manuscript.

11. Abstract:

a. Lines 3-5: Consider revising to say, "CuInS₂/ZnS quantum dot (QD) films were used to down-convert ultraviolet/blue photons to red emissions at 600 and 660 nm resulting in increased biomass accumulation in red romaine lettuce." Reads better.

Revised as suggested.

b. Line 5 (second sentence): Revise to say, "All plant growth parameters except for spectral quality, were held constant..."

Revised as suggested.

c. Line 10: Revise to say, "enhance productivity in greenhouses on Earth, or in space where further improvements are expected..."

Revised as suggested.

12. Introduction:

a. You are missing an Introduction heading at the top of page 2.

The heading was added.

b. 2nd paragraph, line 2: After "crewed space applications," consider inserting the citations from the Zabel et al. 2016 and Wheeler, 2017 references below:

Zabel, P., Bamsey, M., Schubert D., & Tajmar, M. Review and analysis of over 40 years of space plant growth systems. *Life Sciences in Space Research*. 10, 1–16 (2016).

Wheeler, R.M. Agriculture for Space: People and Places Paving the Way. *Open Agriculture*. 2, 14-22 (2017).

References have been included.

c. 4th paragraph, line 1: The use of terms such as "new", "novel", or "for the first time" should be avoided. I suggest replacing "A new technology" with "Here, we present an approach" that uses luminescent...etc.

The novelty is in your work, and it speaks for itself.

Thank you for the comment. We have modified the language accordingly.

13. Results:

a. I recommend this section be designated as a "Results and Discussion" section. I find the sections entitled, "Spectral Quality on Plant Growth," and "Spectral Control with CIS/ZnS QD Films" to be highly introductory in nature and contain a lot of background, and not appropriate to be presented in a Results section.

Section designation was changed as suggested.

b. I recommend the entire "Spectral Quality on Plant Growth" section as is (remove subheading title), be moved and inserted after the 2nd paragraph of the Introduction, and remove the reference to Figure 1a (photosynthetic action spectrum) in the second paragraph. The citations of McCree's papers are sufficient here since you refer to this figure again later on (page 5).

Section was moved and heading was removed as suggested. Figure 1a reference was removed.

c. In the 1st paragraph of "Spectral Quality on Plant Growth," lines 4-6, this sentence should be rearranged. I understand what you are trying to communicate, but the presentation of these fundamental concepts seems out of order. I suggest revising to the following, "The PPFD a plant receives in a 24-hour day is measured as the daily light integral (DLI, mol m⁻² d⁻¹), and is directly proportional to the rate of plant growth which is subsequently determined by biomass accumulation."

Sentence revised as suggested.

d. I recommend the 1st paragraph of the "Spectral Control with CIS/ZnS QD Films" section be moved and inserted after the 3rd paragraph of the Introduction. This paragraph presents a solid background on QD Films, but it is in the wrong place. I believe if you insert it after discussing the gaps/challenges to controlling light quality, it would serve as an effective connecting paragraph to the final Introduction paragraph that summarizes your approach and what you did.

Paragraph rearranged as suggested.

e. Consider starting the Results and Discussion with the 2nd paragraph of the "Spectral Control with CIS/ZnS QD Films" section. However, the entire paragraph 4 and the first sentence of paragraph 5 should be moved to the Methods section.

The indicated paragraphs have been rearranged as suggested with additional minor edits for flow.

f. All statistical analyses appear to be appropriate for the experimental design, but the details should probably be presented in the Methods.

Descriptions of statistical analyses were moved to a subsection for statistics in the Methods section.

g. There is an additional aspect you could mention in the "QD Films Applications for Space" section. Your approach could be also employed during space transit as well. Either en route to Mars, in cis-lunar space, or even aboard the ISS. You may want to mention the potential application of your QD films as coatings for lighting arrays that could be connected to solar concentrators via optical waveguides that could transmit solar light directly into growth chambers, and modifying the spectrum as the final step. For reference, see Nakamura et al. 2010 and Nakamura et al 2012 below.

Nakamura, T., Van Pelt, A.D., Yorio, N.C., Drysdale, A.E., Wheeler, R.M., & Sager, J.C. Transmission and Distribution of Photosynthetically Active Radiation (PAR) from Solar and Electric Light Sources. *Habitation*, 12, 103-117 (2010).

Nakamura, T., & Smith, B.K. Space Solar Energy System for Thermal and Photosynthetic Applications. *American Institute of Aeronautics and Astronautics Space*, 5168, 1-13 (2012).

Also see report by Mickens et al. 2019 below that mentions potential use of harvesting sunlight directly for growth chambers on space modules during transit. This group also used an artificial sunlight fixture comprised entirely of LEDs that could serve as alternative light source above the QD films of your experiment.

Mickens, M.A., Torralba, M., Robinson, S.A., Spencer, L.E., Romeyn, M.W., Massa, G.D., & Wheeler, R.M. Growth of red pak choi under red and blue, supplemented white, and artificial sunlight provided by LEDs. *Scientia Horticulturae*, 245, 200-209 (2019).

Thank you for the recommendations. Indeed, coupling solar concentrators with optical fibers is part of our ongoing research with NASA and we planned to include these references with future published work that is more focused on the topic of light conversion plus waveguiding. However, we agree they are relevant for this publication and so the references have been added along with denotation of this possible use-case.

14. Discussion:

This section should not be designated as a stand-alone Discussion. A discussion consists of an extended analysis of the results and their comparison to the literature rather than a short summary or conclusion. Although this paragraph is well-written and suitable as a summary, I suggest making it the final paragraph of the previous section, "QD Films for Space Applications" for closure.

The discussion section has been incorporated into the final paragraph of the previous section.

15. Methods:

All aspects of the methodology are appropriate with sufficient detail for replication. However, I do suggest a "Statistics and Reproducibility" section be inserted somewhere in this section. A lot of statistical analysis details were presented back in the Results section entitled, "Spectral effects on red romaine lettuce production," and perhaps should be moved here.

A Statistics and Reproducibility subsection was added within the Methods section, and the descriptions of the statistical analyses were moved to this subsection.

Reviewer #3 (Remarks to the Author):

This paper reports on the enhancement of the greenhouse crop production which has been obtained by using a coverage made by a polymeric film doped with luminescent quantum dots (QDs). These latter absorb a fraction of the solar UV spectrum and re-emit light in the red spectral region which is especially beneficial for the growth of many vegetables. The main claim of the paper is that:

“no viable options were available for controlling the quality of light for greenhouse ...”.

However, this claim is completely false. Indeed, there are plenty of examples of films doped with different dyes which are able to provide this same enhancement. Just as an example you can read this paper (the very first one I found in the web) *Europ. J. Hort. Sci.*, 71 (4). S. 145–154, 2006 and the therein references. Furthermore, I am pretty sure that in the 1990s films doped with Europium complexes had even been marketed for this same purpose.

The reported results do not show any peculiar advantage in the use of QDs instead of conventional organic dyes for the light conversion. So, I do not find in the paper any element of novelty that supports the publication of this work in high impact factor journals.

We have modified the manuscript to include past work done on organic fluorophores embedded in greenhouse films as well as work on Europium complexes and clarify the novelty of our work is in utilizing CIS QDs, which solve previous issues with stability and performance of other fluorescent materials. The added references include:

- Rajapakse, N.C.; Young, R.E.; OI, R. Growth responses of chrysanthemum and bell pepper transplants to photoselective plastic films. *Scientia Horticulturae* 84, 215–225 (2000).
- Murakami, K., Cui, H., Kiyota, M., Aiga, I., Yamane, T. Control of plant growth by covering materials for greenhouses which alter the spectral distribution of transmitted light. *Acta Horticulturae* 435, 123-139 (1997).
- Hemming, S., Van Os, E.A., Hemming, J., Dieleman, J.A. The Effect of New Developed Fluorescent Greenhouse Films on the Growth of *Fragaria x ananassa* ‘Elsanta’. *European Journal of Horticultural Science* 71(4),145–154 (2006).
- Pogreb et al. Low-density polyethylene films doped with Europium (III) complex: Their properties and applications. *Polymers Advanced Technologies* 15 (7), 414-418 (2004).

While past attempts at altering spectral quality for crop production with other fluorescent materials have been made, no fluorescent agricultural films have been available commercially, presumably due to poor stability or performance limitations of the alternative materials.

For example, agriculture films doped with Europium complexes referred to by the reviewer exhibit low PL QY (<10%), see Pogreb et al. “Low-density polyethylene films doped with Europium (III) complex: Their properties and applications” 15 (7), 414-418, 2004. Additionally, organic fluorophores, like those used in the Hemming et al. (2006) study referenced by the reviewer often suffer from stability issues. Our manuscript highlights CIS QDs illustrates an interesting alternative to achieve luminescent greenhouse films for improved crop production.

REVIEWERS' COMMENTS:

Reviewer #1 (Remarks to the Author):

Overall this is a strong paper and I am very pleased with the corrections that were added to the paper. I believe that all of the changes were addressed and support publication.

Reviewer #2 (Remarks to the Author):

After review of the author's rebuttal letter, I understand the experimental lighting conditions better, and I agree, the intention is to mimic how the QD films would behave in a greenhouse setting, and the risk of increasing number of lamps or reducing light-to-canopy distance could influence the other treatments.

The authors have made concentrated efforts to implement my recommendations, and I am satisfied with all revisions.